# A Convolutional Neural Network for Classifying Cloud Particles Recorded by Imaging Probes

Georgios Touloupas[1,*], Annika Lauber[2,*], Jan Henneberger[2], Alexander Beck[2], and Aurélien Lucchi[1]

[1]ETH Zurich, Institute of Machine Learning
[2]ETH Zurich, Institute for Atmospheric and Climate Science
[*]These authors contributed equally to this work

**Correspondence:** Annika Lauber (annika.lauber@env.ethz.ch), Jan Henneberger (jan.henneberger@env.ethz.ch)

**Abstract.**

During typical field campaigns, millions of cloud particle images are captured with imaging probes. Our interest lies in classifying these particles in order to compute the statistics needed for understanding clouds. Given the large volume of collected data, this raises the need for an automated classification approach. Traditional classification methods that require extracting features manually (e.g. decision trees and support vector machines) show reasonable performance when trained and tested on data coming from a unique dataset. However, they often have difficulties to generalize to test sets coming from other datasets where the distribution of the features might be significantly different. In practice, we found that for holographic imagers each new dataset requires labeling a huge amount of data by hand using those methods. Convolutional neural networks have the potential to overcome this problem due to their ability to learn complex non-linear models directly from the images instead of pre-engineered features, as well as by relying on powerful regularization techniques. We show empirically that a convolutional neural network trained on cloud particles from holographic imagers generalizes well to unseen datasets. Moreover, fine-tuning the same network with a small number (256) of training images improves the classification accuracy. Thus, the automated classification with a convolutional neural network not only reduces the hand-labeling effort for new datasets but is also no longer the main error source for the classification of small particles.

## 1 Introduction

Clouds play an important role in our weather and climate system. Nevertheless, our understanding of microphysical processes, especially in mixed-phase clouds (MPCs), which is a large source of precipitation in the mid-latitudes (Mülmenstädt et al., 2015), is limited (Boucher et al., 2013; Korolev et al., 2017). Global climate models and satellite-based observations show a large spread in the fraction of ice and supercooled liquid in MPCs (McCoy et al., 2016). Phase-resolved in situ observations can constrain the phase-partitioning in MPCs (Baumgardner et al., 2017). Furthermore, phase-resolved observations of MPCs are crucial to improve the understanding of processes like primary and secondary ice production. Yet, in particular, the measurements of ice crystals below 100 μm remain a challenge (Baumgardner et al., 2017).

For single-particle detection, there are two common measurement instruments: imaging and light scattering probes. The latter (e.g. SID, Cotton et al. (2010), BCP, Beswick et al. (2014), CAS, Baumgardner et al. (2001)) capture the scattered light

of a single particle usually over a range of angles. Applying Mie theory and scale factors, which are derived from calibrations, information of the measured particle like the equivalent optical diameter (EOD) can be derived. However, this can be a major issue for nonspherical ice crystals since the derivation of the EOD assumes sphericity and the exact shape of the captured particle is unknown (Baumgardner et al., 2017).

This issue is partly overcome with imaging probes, which capture images of the particle itself. Assumptions of the shape have only to be made on the third dimension and if the resolution is low compared to the particle size like outlined later in this section. While cloud particle imaging probes like the CPI (Lawson et al., 2001), CIP (Baumgardner et al., 2001) and PHIPS-HALO (Abdelmonem et al., 2011, 2016) directly capture a 2D particle image of single cloud particles, digital in-line holography (e.g., HOLODEC, Fugal and Shaw (2009), HOLIMO 2, Henneberger et al. (2013), HOLIMO 3G, Beck et al. (2017)

and HALOHolo, Schlenczek (2018)) captures the information of an ensemble of cloud particles on a so-called hologram. From such a hologram 2D particle images of the observed cloud particles are computationally reconstructed (see section 2). Based on these particle images, both techniques offer information on shape, size, and concentration of cloud particles. The phase of cloud particles can be determined by distinguishing between circular shaped liquid droplets and non-circular shaped ice crystals (Baumgardner et al., 2017). In addition, the particle images can be used to classify different ice crystal habits (e.g.

Lindqvest et al., 2012). The differentiation of liquid droplets and ice crystals by their shapes requires a minimum diameter which depends on the complexity of the ice crystal. To identify needles six pixels might be enough, whereas a minimum of 12 to 15 pixels is required to identify plates while frozen droplets which are spherical cannot be detected regardless of their diameter (Korolev and Sussman, 2000; Korolev et al., 2017).

During a typical field campaign, millions of images of cloud particles are captured. For phase resolved measurements in

MPCs, the cloud particles need to be classified into liquid droplets, ice crystals and in the case of holography also into artifacts, which are usually part of the interference patterns of larger particles (more details in section 2). The classification of such a large amount of images by hand is a very time-consuming step, which raises the need for an automated classification algorithm. Because ice crystals are very variable in shape and their typical maximum dimensions of about 1-1500 µm are overlapping with the size range of cloud and rain droplets (about 1 µm to a few millimeters) together with artifacts, which can have any shape

and size, it is difficult to develop a classification algorithm to perform well.

Imaging probes, which differentiate only ice from liquid, usually extract features from the images that measure the circularity of the particles (e.g. Korolev and Sussman (2000); Crosier et al. (2011); Lawson et al. (2001)). Korolev and Sussman (2000) state an uncertainty for differentiating spheres from irregular particles of 20% to 25% for a pixel number between 20 and 60 and a few percents for higher pixel numbers. These values are comparable to our results for holographic images, which we

will introduce later. However, the existing approaches are not suitable for holographic images since they do not account for artifacts. Finding good features for artifacts is difficult because they do not have a specific shape.

Therefore, classification methods that are commonly used for holographic imagers rely on supervised machine learning approaches which learn to predict labels (a.k.a. classes) from labeled training data. The algorithm is trained with input samples, called training data, which were labeled by humans (i.e. hand-labeled) into a defined number of classes. For example, the

images can be divided into three classes, namely liquid droplet, ice crystal, and artifact. Many supervised machine learning

algorithms cannot directly handle particle images as inputs, but instead require extracting a set of features from the images (e.g. sphericity, area, etc.). The performance of the algorithm describes how well these features can distinguish between the different classes. For instance, decision trees and support vector machines (SVMs), which were used for the classification of ice crystal shapes by Grazioli et al. (2014) and Bernauer et al. (2016) use around 10 to 30 extracted features as input. Although these algorithms might achieve reasonable results when trained and tested on a single dataset, they often have difficulties to generalize to different datasets whose features might follow a slightly different distribution. This problem is commonly referred to as transfer learning in the machine learning community (see details in Goodfellow et al. (2016)).

Deep learning (usually referred to as neural networks) has the potential to overcome transfer learning issues, which we will show in this work. For the classification of cloud particles, a feedforward neural network from Hagan and Menhaj (1994) was used by O'Shea et al. (2016) to classify CPI data into different ice particle shapes and liquid droplets. The network is fed by different features, which are calculated beforehand. Their results are promising with a total accuracy of 88% to classify the images into six habits including liquid droplets for particles larger than 50 μm. This type of a neural network also requires feature extraction and does not work for holographic images because it does not account for a class without a specific shape like artifacts.

In this paper, we suggest using a convolutional neural network instead, which does not require any manual feature extraction but can instead use the particle images as inputs directly. Automatically extracting features from image data allows for more reliable and robust predictions to be made, which we will demonstrate empirically. Our experimental setup consists in training a convolutional neural network (CNN) on several real-world datasets of cloud particles with major axis sizes larger than 25 μm (see section 2) recorded by different holographic imagers. This technique achieves higher accuracy and exceeds the generalization abilities of a decision tree and SVM baseline (see section 4.4). Nevertheless, we still have to account for a varying accuracy over particle size (see section 4.5). We also introduce a fine-tuning approach that can significantly improve the performance of the CNN (see section 4.2).

## 2 Experimental data

The data used for training and validation of the classification algorithms presented in this paper were obtained in situ with several holographic imagers utilizing digital in-line holography. For this technique, a laser irradiates an ensemble of particles inside a cloud volume. The particles in this cloud volume scatter the laser light and the scattered wavefronts interfere with the unscattered laser beam. The resulting interference pattern (hologram) is captured by a digital camera (see Fig. 1). From the hologram, 2D complex images are numerically reconstructed at several planes within the cloud volume of regular distance (usually 100 μm) using the software HOLOSuite (modified version of Fugal et al., 2009) and depicted as amplitude and phase images (see Fig. 2). The software then detects images of the particles in different planes within the reconstructed amplitude images by using an amplitude threshold and patches the images of the particles in adjacent planes which are at the same lateral position. Patched traces are assumed to be one particle. The focus planes of particles within particle patches are determined by

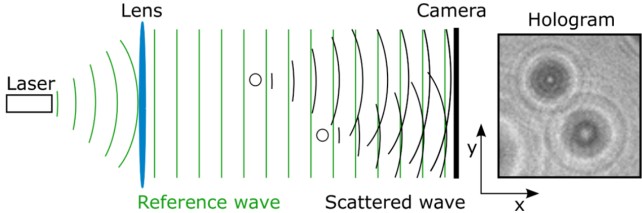

**Figure 1.** Working principle of digital in-line holography. The laser is collimated by a convex lens. Particles between the lens and the camera scatter the light which interferes with the plane reference wave. The interference pattern is captured by the camera as a hologram from which images in different distances to the camera can be reconstructed. Figure adapted from Beck et al. (2017).

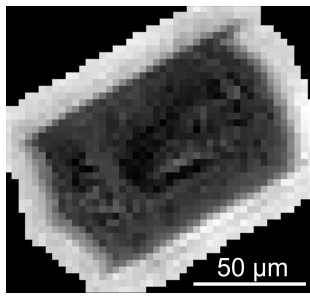 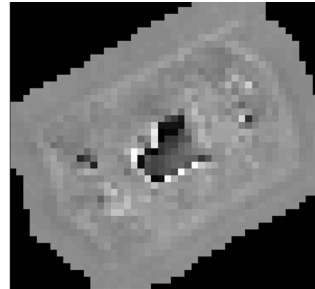

**Figure 2.** Channels computed from an ice crystal complex image. **Left:** Amplitude channel **Right:** Phase channel

using an edge sharpness algorithm. A more detailed description of the measurement technique and software can be found in Henneberger et al. (2013) and Fugal et al. (2009).

To reduce the amount of data, only the particle pixels and a few pixels of their surrounding were saved for the subsequent analyses. The data was hand-labeled into three classes: liquid droplets (circular particles), ice crystals (non-circular particles) and artifacts (parts of the interference pattern, scratches on the windows, noise, etc). The decision was usually made based on the amplitude images in the focus and its neighboring planes but also the traces of the amplitude images within a particle patch (like the gradients of the maximum and minimum values) provide useful information. Traces of artifacts are rather fluctuating while traces of particles show a clear maximum. For training and testing the CNN, only the amplitude and phase images of the focus plane were used.

Five different hand-labeled datasets from two different instruments (HOLIMO 3M and HOLIMO 3G (Beck et al., 2017)) were used for testing and training. The datasets were obtained at different measurement sites in different weather situations and were labeled by different people (Table 1).

The class distributions vary strongly between the datasets which is due to different environmental conditions but also to different preprocessing routines of the data (Table 1). For example, a lower amplitude threshold for particle detection during the reconstruction results in a larger fraction of artifacts. Furthermore, the more particles are inside a measurement volume, the lower is the signal-to-noise ratio and the higher the number of artifacts. The pixel intensity distributions of the datasets give an

**Table 1.** Detailed information on the different datasets used for training and testing the CNN.

| Dataset | Instrument | Location | Date | Total (#) | Liquid droplets (#) | Ice crystals (#) | Artifacts (#) | Labeled by |
|---|---|---|---|---|---|---|---|---|
| iHOLIMO 3G | HOLIMO 3G | Jungfraujoch, Switzerland | 11/2016 | 22373 | 17547 | 894 | 3932 | Annika Lauber |
| iHOLIMO 3M | HOLIMO 3M | Jungfraujoch, Switzerland | 11/2016 | 20253 | 4166 | 393 | 15694 | Sarah Barr |
| JFJ 2016 | HOLIMO 3G | Jungfraujoch, Switzerland | 03/2016 | 7221 | 1744 | 516 | 4961 | Jan Henneberger |
| SON 2016 | HOLIMO 3G | Hoher Sonnblick, Austria | 03/2016 | 15648 | 7056 | 2215 | 6377 | Alexander Beck |
| SON 2017 | HOLIMO 3G | Hoher Sonnblick, Austria | 02/2017 | 19476 | 151 | 17453 | 1872 | Alexander Beck |

indication of their different noise levels. Because the distributions differ substantially between the datasets (see Appendix B), it has been difficult to develop an automated classification algorithm working for all of them.

The classes predicted by the CNN were evaluated against the hand-labeled classes, which are considered as the ground truth. Nevertheless, some particles in the training datasets can also have a wrong class/label. Because the classification was done by humans, misclassification can have happened by accidentally pressing the wrong button or by misinterpretation of the image. In some cases, it is not possible to distinguish between two classes, e.g. circular or very small ice crystals cannot be separated from liquid droplets. In addition, optical distortion compromises the classification. Particles far away from the camera have a blurry edge as the resolution is getting worse and at the edges, the lens distortion deforms circular particle to an oval shape. The training dataset was constrained to particles with a major axis size larger than 25 µm, corresponding to 8 pixels, which is the empirical threshold from which a reliable determination of the classes for most of the particles is possible.

For the estimation of the human bias, three different people hand-labeled the same dataset consisting of 1000 particles. The number of particles hand-labeled as the considered class by at least one person are compared to the number of particles hand-labeled as the considered class by all three persons. Taking the average of these two numbers, the spread can be given as the percentage deviation to the two values (see Fig. 3). For liquid droplets, we have a deviation of $\pm 4\%$ and for ice crystal $\pm 5\%$. However, this estimation does not take into account that in some cases humans might just not be able to recognize the correct class as outlined before.

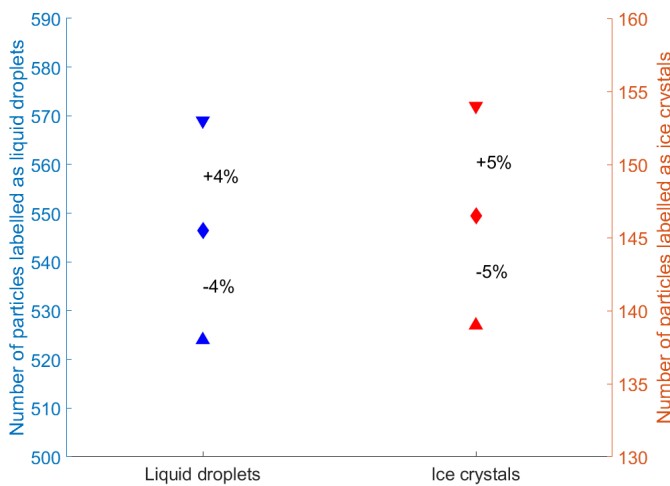

**Figure 3.** Evaluation of the human classification bias from three people labeling the same dataset consisting of 1000 particles. The upward pointing triangles are the total number of particles labeled by all three people as the considered class (100% agreement). The downward pointing triangles are the total number of particles classified by at least one person as the considered class ($\geq$33% agreement). The diamond is the average between the two points with the percentage values as the deviation to them.

## 3 Network architecture, implementation and evaluation method

In this section, we discuss the architecture of the CNN used in our experiments and provide details about the training procedure as well as the fine-tuning method. We also introduce the baselines used for comparison. We refer readers unfamiliar with deep learning to Appendix A where they can find a general introduction about deep CNNs.

### 3.1 Network architecture

The CNN used during this work was adapted from the VGG (Visual Geometry Group) architecture of Simonyan and Zisserman (2014) as shown in Fig. 4. The first part of the architecture consists of multiple convolutional blocks, each having two or three convolutional layers followed by a max-pooling layer. The architecture was adjusted to accept $32 \times 32 \times 2$ pixels (amplitude and phase images) as inputs. For the convolutional part of the CNN, four convolutional blocks are used, the first two containing

10 two convolutional layers and the next two containing three convolutional layers. The convolutional layers use $3 \times 3$ filters with stride $S = 1$, zero-padding and ReLUs (Rectified linear units) as the activation functions. The number of filters is the same for all convolutional layers inside the same convolutional block and increases throughout the convolutional part of the network, from 64 to 128, 256 and eventually 512. The pooling layers perform max-pooling with stride $S = 2$, each one reducing the spatial dimensions by two. In total, 2048 features are extracted in the convolutional blocks.

For the classifier part of the network, three fully connected layers progressively reduce the dimensions from 1024 to 128 to 3 (artifact, liquid droplet, ice crystal). A ReLU is also used as the activation function for the fully connected layers.

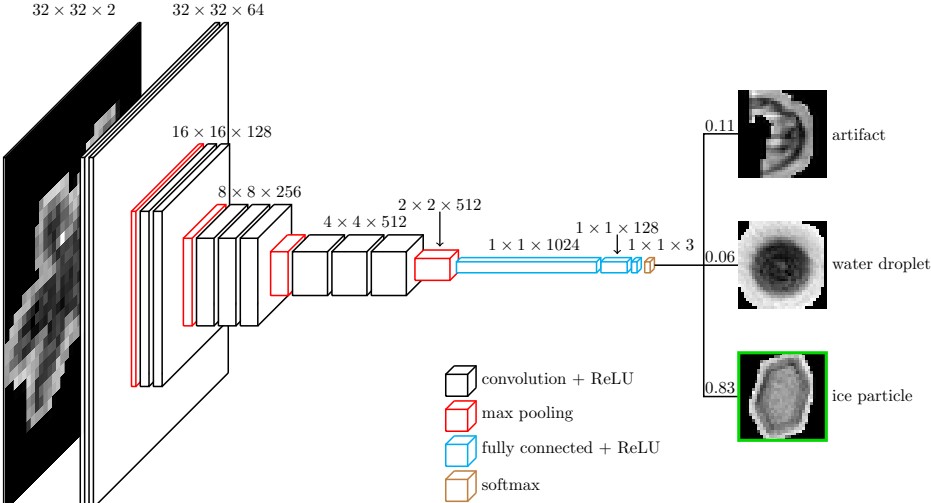

**Figure 4.** The VGG-based CNN architecture. A preprocessed image of canonical size $32 \times 32 \times 2$ is fed to the CNN architecture which consists of a sequence of layers that implement various operations (see the detailed description of each operation in the Appendix C). The CNN processes the input image and outputs probabilities for each of the three classes considered here. The class with the highest probability is then chosen as the prediction (the class with the green border in this Figure, having a softmax probability of 0.83). Figure adapted from Cord (2018).

To speed up training, dense residual connections are used (Huang et al., 2017). Inside each convolutional block, connections from every layer to all the following layers were added. Similarly to He et al. (2016), batch normalization is also performed right after each convolutional and fully connected layer and before the ReLU activations, as well as after the pooling layers and the input layer.

## 3.2 Data preprocessing

*Normalizing the input*

The images were preprocessed to a standard format before being given to the CNN (Fig. 5). Standard CNN architectures (including VGG) require a fixed square size image as input. The size of the input image is a design choice that is typically determined by various restrictions. While larger images slow down training and increase the amount of memory required by the network, smaller input images reduce the resolution and therefore the information available in the images. We empirically found that using an image size of $32 \times 32$ pixels leads to satisfying results. These images were created as follows. From the raw complex image data (see Section 2), both the amplitude and phase images were extracted. Square images were created by filling the unknown pixel values with black pixels (zero-padding) and scaling the images to $32 \times 32$ pixels. The amplitude and phase images were fed to the CNN as two different channels.

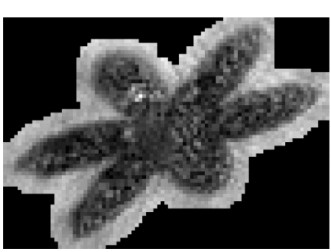 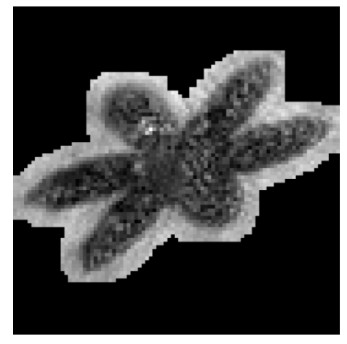 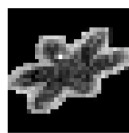

**Figure 5.** Preprocessing steps for an ice crystal image (only the amplitude channel is shown). **Left:** Original $89 \times 63$ image **Center:** Zero-padded $89 \times 89$ square image **Right:** Scaled $32 \times 32$ image used as input to the network.

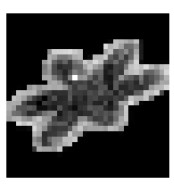 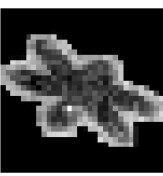 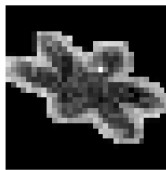 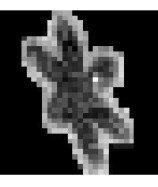 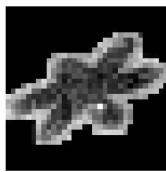

**Figure 6.** Data augmentation transformations applied on an ice crystal image, visualization of the amplitude channel. The transformations are applied to the original image. **Left to right:** original image, vertical flip, horizontal flip, 90° rotation, transposition.

*Data augmentation*

Techniques such as data augmentation help a CNN to achieve lower generalization errors (measure of how accurately an algorithm is able to predict outcome values for previously unseen data), especially when the amount of data available for training is limited. We augmented the training data by performing transformations that preserve the shape of the particles such as vertical flip, horizontal flip, 90° rotation and transposition (see Fig. 6). Each transformation was applied with a 50% probability to every input image before it was fed to the network during training. This means that multiple transformations could have been applied to one image during training. This process was switched off after training.

## 3.3 Training details

The weight parameters of the network were initialized using the Xavier initialization scheme (Glorot and Bengio, 2010), while all biases were initialized to $0.1$. The loss function (see Appendix A2) used to train the network is a softmax cross-entropy which was modified to combat the issue of having imbalanced classes. This problem arises when a dataset contains significantly more labels of one class compared to the other classes. Since the cross-entropy loss penalizes the misclassification of all classes equally, the learned model will be biased towards the class with the highest class frequency. To avoid this behavior, the importance of each class in the cross-entropy loss was weighted by its inverse class frequency. This re-weighting results in an increase in performance for the rarer classes. In order to optimize the weights of the network, a variant of stochastic

gradient descent named Adam (Kingma and Ba, 2014) was used. The learning rate was set to $\eta = 10^{-4}$ and the gradients were computed using mini-batches of size 256 samples.

In order to prevent overfitting to the training set which would increase the generalization error, a separate validation set was used to select the best set of parameters. This set was created by splitting the data available for training (not including data for testing) into a training set and a validation set. The training set was used for training the network as described so far, whereas the validation set was used to evaluate the performance of the network on unseen data. Typically the loss of the network for the training set decreases steadily during training, while the loss on the validation set will start to increase after a certain point. This is a sign that the network starts to overfit to the training set and therefore the training was stopped (this method is referred to as early stopping). In our implementation, the loss of the validation set was computed after each epoch (one pass over the training set). Every time the loss decreased, the current parameters were retained. If the validation loss did not improved during the last 10 epochs, the training was stopped.

In the first part of the experiments, the model was trained using data from a single dataset. Therefore, the datasets were randomly divided into a training, a validation and a test set with a 60-20-20 split, which means that we used 60% of the dataset as training, 20% as validation and 20% as test data. The same models trained in this way on a single dataset were also tested on all other available single datasets. In the second part of the experiments, the model was trained using data from merged datasets and tested on a different one. In order to train the network, the merged dataset was divided into a training and a validation set with a 90-10 split since the merged dataset contains more data than the single dataset used in the previous experiments and the test set is an unseen dataset in this case.

### 3.4 Fine-tuning

Using the CNN trained on a dataset with different characteristics from the target dataset may lead to less good or even poor results. A fine-tuning approach that reuses the weights of the pre-trained CNN as initialization for a model being trained on a sample of the target set may overcome this issue. This approach requires only a low number of training samples to be powerful (a few hundred as explained later), which speeds up training compared to training a model from scratch.

In our experiments, we fine-tuned the weights of the CNN using samples sizes between 32 and 2048. Each sample set was divided into a training and a validation set with an 80-20 split and the data not used as sample set were used as the test set, on which the model was evaluated after fine-tuning. A smaller learning rate of $\eta = 10^{-4}$ was used for fine-tuning. Since the training sets used for fine-tuning are much smaller than the full datasets, we decreased the mini-batch size to 4/8/16/32/64/128/128 for sample size 32/64/128/256/512/1024/2048 and increased the number of epochs without loss improvement before early stopping to 1600/800/400/200/100/50/50 for sample size 32/64/128/256/512/1024/2048.

### 3.5 Baseline of classification algorithms used for comparison

We compared the performance of the CNN to two other supervised machine learning techniques (a decision tree and a SVM), all of them being trained and tested on the same datasets. The details of these baselines are described below.

### 3.5.1 Classification tree

A classification tree is a common classification method in machine learning that makes decisions about a given input image by performing a series of hierarchical binary tests on the features extracted from the input. This hierarchical structure can be thought of as a tree where each node in the tree is a logical binary decision on one selected attribute, e.g. checking whether the diameter of a particle is larger or smaller than 50 μm. Each input sample begins at the root of the tree and ends at a leaf node which is associated with one of the predefined classes. More information about classification trees can be found in Breiman (1984).

We created our decision trees using the Matlab function *ClassificationTree.fit* with a minimum leaf size of 40 (each leaf has to include at least 40 sample points of the training data). For training, we extracted 32 features (see Appendix G1) for every particle in the training set. The datasets were split into training, validation, and test sets according to the experiments with the CNN. The validation set was used to find the best pruning level. We also added a cost function that penalizes the misclassification of a particle by its inverse class frequency to avoid a bias towards the largest class.

### 3.5.2 Support vector machine

Support Vector Machines (SVMs) are a standard supervised classification technique in machine learning. For each object in an image, they require a set of feature vectors as well as a corresponding label. Standard SVM training algorithms find a hyperplane separating each pair of classes so that the distance between the closest data point and the hyperplane (margin) is maximized. This procedure leads to an optimization problem that can be solved using a Quadratic Programming (QP) solver.

In the results presented below, each SVM was trained on the same set of features as the decision tree (see Appendix G1). The datasets were split into training, validation and test sets according to the experiments with the CNN and the validation set was used to find the best hyperparameters. We also used a radial basis function (RBF) kernel to increase the discriminative power of the features (Hsu et al., 2003).

### 3.6 Evaluation metrics

For the evaluation of the results different metrics were used, each of them measuring different characteristics of interest. For classes with very few samples, it may be important to detect all particles belonging to it (high accuracy) but also not to overestimate the frequency of the class by mistakenly classifying a few percentages of a large class as being part of the small class (low false discovery rate). In order to mitigate this problem, we make use of a diverse set of evaluation metrics which are presented below.

*Confusion matrix*

One way to summarize the performance of a classification model is the confusion matrix. In a confusion matrix $C$, each element $C_{i,j}$ is equal to the number of particles labeled as class $i$ but predicted as class $j$. The correct predictions are located on

the diagonal of the matrix, while the classification errors are represented by the elements outside the diagonal.

*Overall accuracy*

The simplest metric to evaluate the performance of a classification model is the overall accuracy, which is defined as the ratio
of the number of correct predictions to the total number of particles. By using the elements $C_{i,j}$ of the confusion matrix, the
overall accuracy $OA$ can be computed in the case of $K$ classes as

$$OA = \frac{\sum_{i,j=1, i=j}^{K} C_{i,j}}{\sum_{i=1}^{K} \sum_{j=1}^{K} C_{i,j}} \tag{1}$$

*Overall FDR*

The overall false discovery rate ($FDR$) is defined as the ratio of the numbers of false predictions to the total number of particles
and is, therefore, equivalent to $1 - OA$.

*Per-class accuracy*

A shortcoming of the overall accuracy metric is that it does not show how well the classification model performs for the
individual classes. If the classes of the test set are imbalanced, the overall accuracy will give a very distorted picture of the
performance of a classifier since the class with the highest frequency will have a dominating effect on the computed statistics.
In this case, it is useful to compute the per-class accuracies to evaluate the performance of the model separately for each
class. The accuracy of a class is defined as the ratio of the number of the correct predictions for this class to the number of
particles labeled as the same class. By using the elements $C_{i,j}$ of the confusion matrix, the per class accuracy $ACC_k$ for class
$k = 1, ..., K$ can be computed as

$$ACC_k = \begin{cases} \frac{C_{k,k}}{\sum_{j=1}^{K} C_{k,j}}, & \text{if } \sum_{j=1}^{K} C_{k,j} \neq 0 \\ 1, & \text{otherwise} \end{cases} \tag{2}$$

*Per-class $FDR$*

The per-class $FDR$ is the ratio of false predictions to the total number of predictions of a class. It can take values between
0% (no particle was wrongly predicted as the considered class) and 100% (all predicted particles do not belong to the con-
sidered class). It is especially important for low frequency classes where a few extra particles can lead to a relatively high
overestimation. The per-class $FDR$ is $FDR_k$ and can be computed as follows for the classes $k = 1, ..., K$

$$FDR_k = \begin{cases} 1 - \frac{C_{k,k}}{\sum_{i=1}^{K} C_{i,k}}, & \text{if } \sum_{i=1}^{K} C_{i,k} \neq 0 \\ 0, & \text{otherwise} \end{cases} \tag{3}$$

*Deviation from ground truth*

The deviation from ground truth (DGT) of a specific class is the absolute number concentration of particles automatically

classified as the considered class to the manually classified (ground truth) number of particles of the same class. Mathematically, one has to subtract the percentage of mislabeled particles described by the FDR value and add the missing particles described by 1-accuracy. The calculation can be applied accordingly for the overall ground truth with the overall FDR and accuracy. This metric is important if only the total number concentration is of interest and not the class of the single particles, e.g. if 100 ice particles are classified as liquid droplets and 100 liquid droplets are classified as ice particles, the deviation from ground truth is zero.

$$DGT_k = 1 - (1 - FDR_k) \cdot (2 - ACC_k) \tag{4}$$

## 4  Results

In this section, the prediction performance ability of the CNN being trained and tested on a single dataset and also tested on other single datasets (generalization ability) is demonstrated (section 4.3). In a third experiment, the generalization ability of the CNN being trained on merged datasets (higher quantity of more diverse data) and tested on an independent test dataset was tested (section 4.4). All three experiments are compared to the results of a decision tree and an SVM approach (details of these approaches in section 3.5.1 and 3.5.2). Moreover, it is shown that using the phase images on top of the amplitude images improves the performance of the CNN (section 4.1) and how fine-tuning the CNN improves its generalization ability (section 4.2, 4.3, 4.4 and 4.5). For being able to evaluate the uncertainty of the mass concentration, we also look at the prediction performance of the CNN over the particle size (section 4.5).

### 4.1  Input channels

As described in Section 3.2, the input channels fed to the CNN were the amplitude and phase images of each particle. Since the phase channel did usually not help to classify the images by hand, it was tested if the prediction ability of the CNN changed by turning the phase channel on and off. Therefore, the iHOLIMO 3G dataset was trained and tested with and without the phase images in three runs (Fig. 7). All evaluation metrics improved using both channels, which was especially pronounced in the per-class accuracies and $FDRs$ of artifacts and ice crystals, with an improvement of about 10% regarding the median of the three runs. Therefore, we conclude that the phase images provide useful information which is not visible to the human eye. Furthermore, the results of the experiments where the phase channel was turned on showed more robust results (less variation between the different runs of the CNN). This is in particular important if the CNN can be trained only once because more runs are computationally more expensive.

### 4.2  Fine-tuning

To evaluate the power of fine-tuning of the pre-trained CNN to a new dataset, the fined-tuned CNN is compared to the CNN trained from scratch using different sample sizes. Both methods are compared to the results of the model trained on the

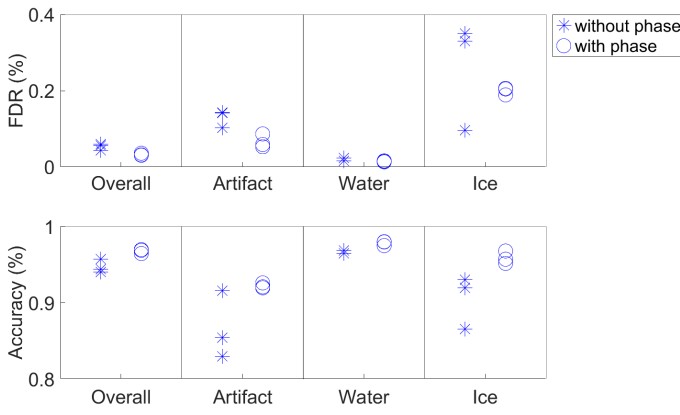

**Figure 7.** Evaluation metrics comparing the CNN trained with the amplitude channel only and trained with both the amplitude and phase channels. Three runs of the CNN being trained and tested on the iHOLIMO 3G dataset are shown.

merged datasets without fine-tuning being applied. Only the results of the iHOLIMO 3G dataset are discussed as the other four experiments showed similar results.

Training the CNN with 2048 samples from scratch achieved overall better results than applying the trained CNN from the merged datasets (Fig. 8). An improvement of the FDR of ice crystals was already observed with 512 samples. This is probably
due to the fact that the relative amount of ice crystals varies the most between the different datasets compared to the other classes. The more different the distributions of datasets are, the harder it is to find an algorithm working for all of them.

Fine-tuning was performing better than training from scratch over all shown sample sizes. Fine-tuning with 256 or more samples outperformed the initial models. Since the overall accuracy did not obviously improve for a higher number of samples, it seems that using 256 samples is a good tradeoff between hand-labeling effort and performance gain. In the following
comparisons we therefore always use the fine-tuning results for a sample size of 256.

### 4.3   Single dataset experiments

A model was trained using data from a single dataset and tested on separate data from the same single dataset (Fig. 9) and on the four other single datasets (Fig. 10). The prediction performance of three classification algorithms (decision trees, SVMs and CNNs) are compared.
When trained and tested on data extracted from the same single dataset, the CNN, decision tree, and SVM all yield median values above 90% for the overall accuracy and the per-class accuracies and below 10% for the per-class FDRs (Fig. 9). The interquartile range of the decision tree is remarkable larger for the accuracy of ice and the FDR of water compared to the SVM and the CNN. Apart from that, all three methods yield similar values.

When testing the same models on the other four datasets (which were not used for training) all metrics worsened indepen-
dently of the classification algorithm being used (Fig. 10). The medians of the overall accuracies and the per-class accuracies

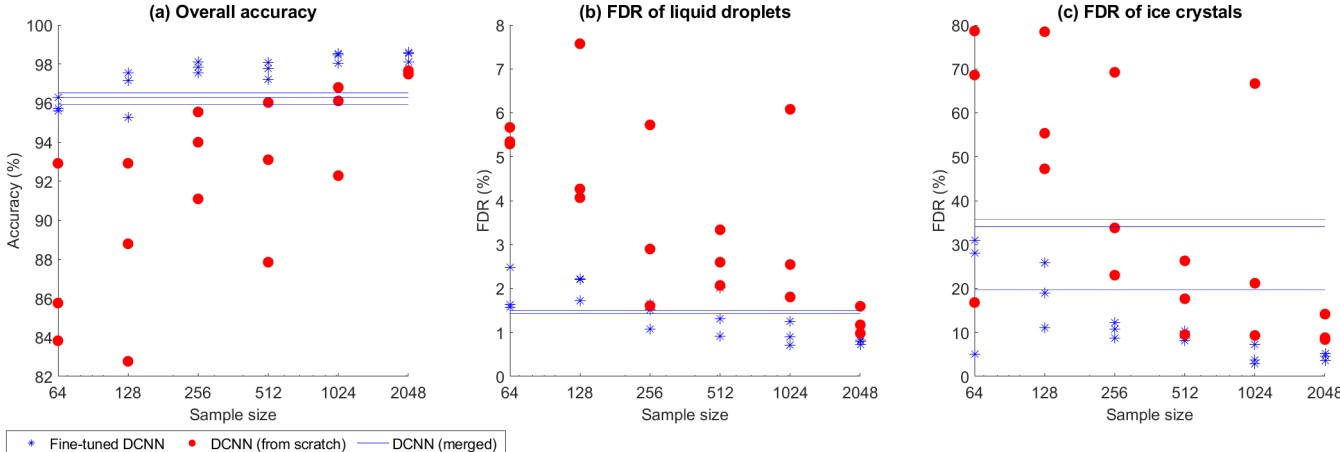

**Figure 8.** Evaluation metrics for predicting the iHOLIMO 3G dataset. The blue lines show the CNNs trained on the other four datasets. A sample of the iHOLIMO 3G dataset was used to either fine-tune these CNNs (green asterisks) or train the CNNs from scratch (red crosses). For each sample size, three runs were performed. If not all classes were present in the training and validation sets due to the limited sample size, the results are excluded.

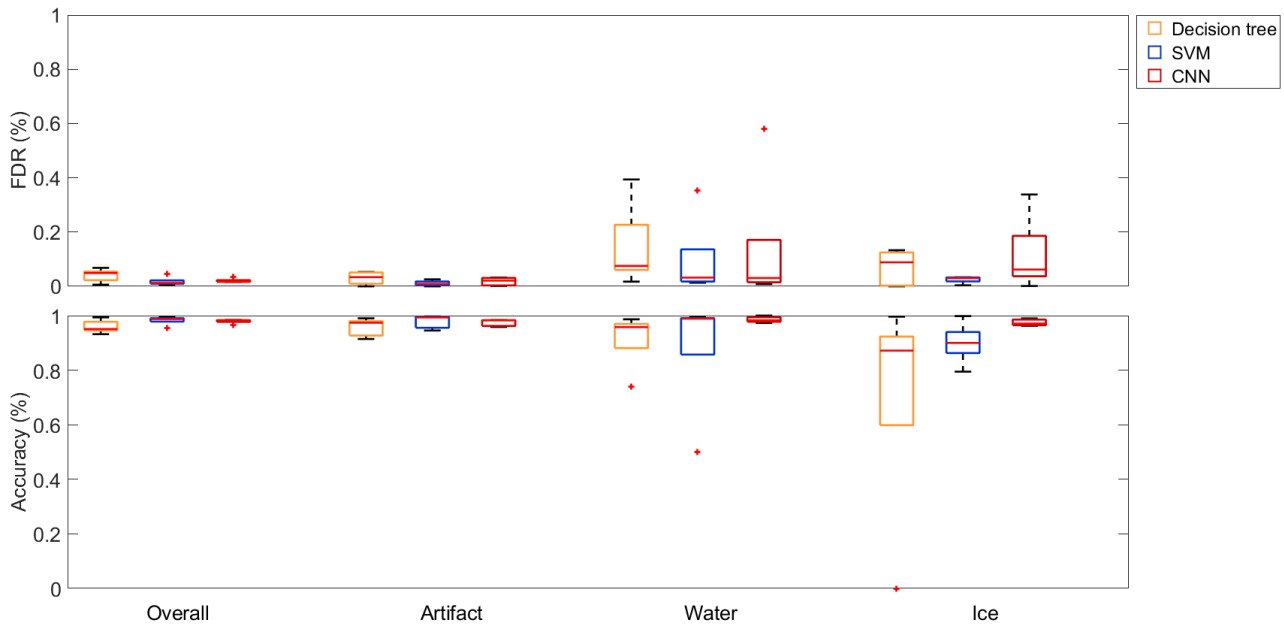

**Figure 9.** Box plots of the evaluation metrics of different classification algorithms trained and tested on the same single dataset. Shown are the results of the decision tree, the SVM, and the CNN being trained and tested on the five datasets. The red lines in the boxes mark the medians while the box edges show the 25th and the 75th percentile respectively. The end of the whiskers are the most extreme data points and the red crosses mark outliers which are defined as data points being more than 1.5 times the interquartile range away from the top or bottom of the boxes.

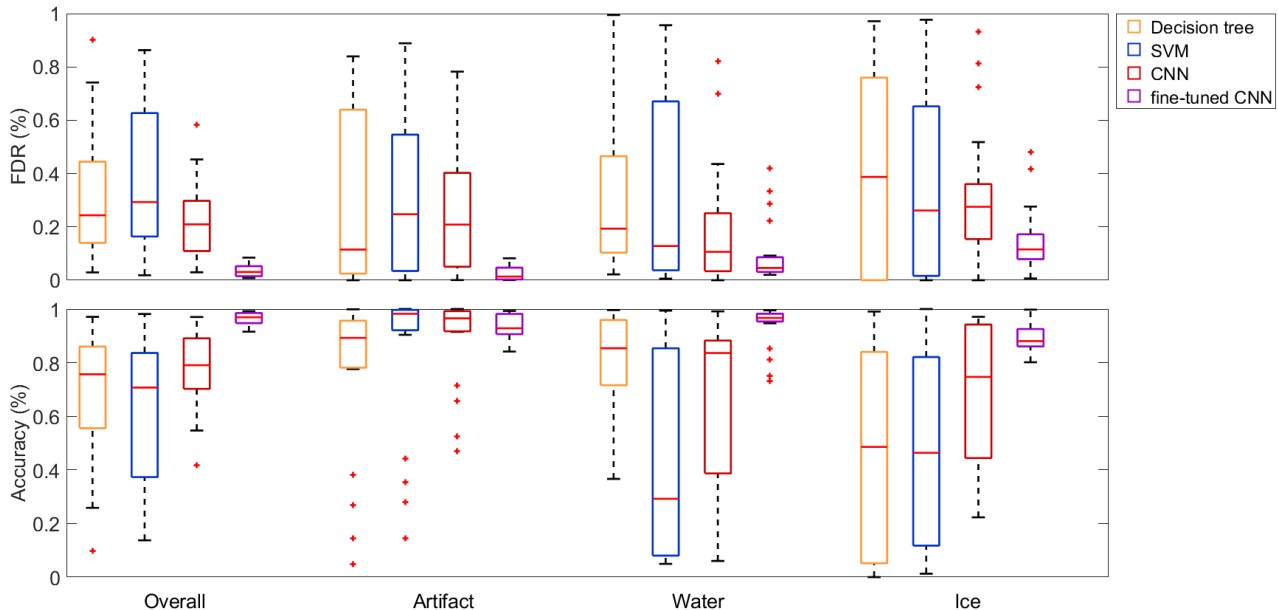

**Figure 10.** Box plots of the evaluation metrics of different classification algorithms trained on a single dataset and tested on another single dataset. Shown are the results of the decision tree, the SVM, the CNN, and the fine-tuned CNN being trained on the five datasets and tested on the other four datasets respectively. The red lines in the boxes mark the medians while the box edges show the 25th and the 75th percentile respectively. The end of the whiskers are the most extreme data points and the red crosses mark outliers which are defined as data points being more than 1.5 times the interquartile range away from the top or bottom of the boxes.

of ice dropped below 80% for all three methods. Only the accuracies of artifacts stayed at relatively high values above 90%. Similarly, all FDR values rose to median values above 10%. Another obvious trend is the larger spread in all metrics, which is likely due to the different per-class distributions of the different datasets, i.e. a dataset trained with mostly ice crystals is not likely to perform well on a dataset trained with mostly liquid droplets but might perform well on a similarly distributed dataset.

5   Nevertheless, the CNN achieved better median values and showed less spread in most metrics and therefore, seems to perform slightly better than the decision tree and SVM in the generalization task.

After fine-tuning the CNN with 256 samples from the target dataset, all metrics (except the accuracy of artifacts) improved. The overall accuracy reached again values above 90% and the per-class accuracies all rose to median values above 80% while the per-class FDRs dropped to values mostly below 20%. Additionally, the maximum and minimum values of the metrics were

10   less spread compared to the algorithms without fine-tuning being applied. Therefore, we conclude that fine-tuning is a very effective method when the model is trained only on a single dataset and used for datasets with different distributions of the classes. We do not exclude that this statement most likely also applies to fine-tuning the decision tree or SVM.

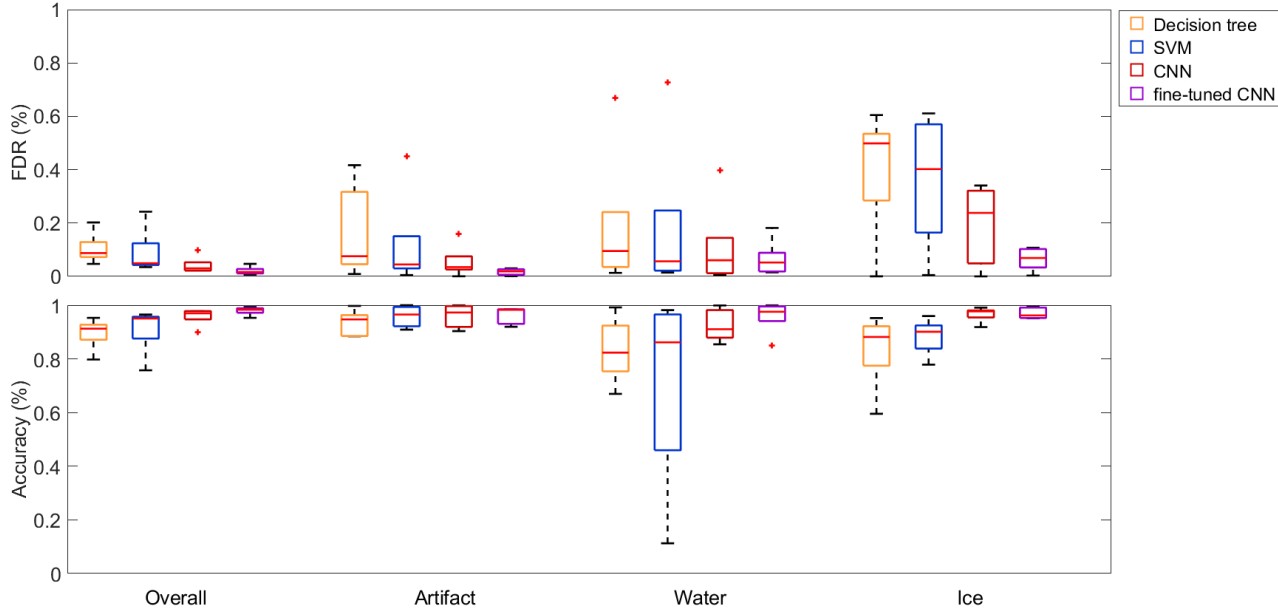

**Figure 11.** Box plots of the evaluation metrics of different classification algorithms trained on four merged datasets and tested on a fifth dataset. Shown are the results of the decision tree (blue), the SVM (green), the CNN (red) and the fine-tuned CNN (purple). The red lines in the boxes mark the medians while the box edges show the 25th and the 75th percentile respectively. The end of the whiskers are the most extreme data points and the red crosses mark outliers which are defined as data points being more than 1.5 times the interquartile range away from the top or bottom of the boxes.

## 4.4 Merged dataset experiments

In order to improve the generalization ability of the CNN, it was trained using data from multiple datasets. We selected the dataset that the network was evaluated on and used data from the other four available datasets to train the network (merged datasets).

5     By using a larger quantity of more diverse data for training, the results greatly improved compared to the single dataset experiments with the median $OA$ rising from values below 80% to above 90% (decision tree: 91%, SVM: 95%, CNN: 97%) for all three models (Fig. 11). Furthermore, the interquartile ranges reduced from values of over 40% to about 25% for the decision tree and the SVM while it reduced from 26% to 15% for the CNN regarding the per-class metrics. All in all, the performance of the CNN surpassed the classification tree and SVM baselines in almost every metric, both in terms of the
10   median values and their spread. Above that, fine-tuning the CNN lead to even better median values (98% median $OA$) as well as smaller interquartile ranges (on average about 5% for the per-class metrics).

## 4.5 The CNN prediction performance over particle size

In order to assess the performance of the CNN with respect to the particle size, the data was split into bins according to the major axis size. Because the small particles outnumber the large particles, an exponentially increasing bin width was chosen. The accuracies and FDRs of liquid droplets and ice crystals for the pre-trained CNN on merged datasets without (left side) and with fine-tuning (right side) are shown in Fig. 12 together with the class distribution of each size bin. Results of size bins where datasets have less than 30 particles of the considered class were excluded due to insufficient statistical representativeness. This is mainly the case for droplets larger than 89 μm.

The accuracy and FDR results without fine-tuning being applied (Figure 12, left side) show a strong variation in particle size. In general, the prediction performance was worse for sizes with a relatively and/or absolutely small amount of samples belonging to the considered class, e.g. for ice crystals smaller than 89 μm (median FDR 20%, median accuracy 93%) and liquid droplets larger than 89 μm (median FDR 25%, median accuracy 79%). For the more common sizes, the values for ice crystals (median FDR 5%, median accuracy 99%) as well as for liquid droplets (median FDR 5%, median accuracy 92%) are better.

Independent of size, the FDR was generally larger for classes which are less frequent in the test dataset. This is expected as the misclassification of only a few percents of a relatively prevalent class leads to a high FDR of a relatively small class. Since ice crystals are scarce compared to liquid droplets in four datasets (see Table 1), their FDR is relatively high for ice compared to liquid (median FDR of ice: 12%, median FDR of liquid 5%), while the opposite holds true for the SON 2017 dataset (median FDR of ice: 0%, median FDR of liquid: 28%) where liquid droplets are underrepresented.

In most cases, fine-tuning led to an obvious improvement in the accuracy and FDR (Fig. 12, right side). There was one outlier for the FDR of ice crystals where one fine-tuning run on the dataset iHOLIMO 3M increased the FDR up to 20%. It is possible that an unusual or even mislabeled example of an ice crystal was in the randomly chosen fine-tuning training sample which has a strong influence on the classification if there are not many other examples. Ignoring this one outlier, fine-tuning of the merged CNN improved the results for the accuracy as well as the FDR over all size bins, e.g. for the common sizes, the median FDR for ice crystals dropped from 5% to 3% while the median accuracy for liquid droplets rose from 92% to 98%. For ice crystals smaller than 89 μm, the median FDR dropped from 20% to 11%.

Due to small numbers of samples, it is difficult to assess the performance of the CNN for liquid droplets larger than 89 μm. Only two datasets contained more than 30 particles for the larger three size bins and the results show that the CNN performed poorly (median FDR 25% and median accuracy 79%) for liquid droplets in those size bins. However, after fine-tuning was applied the median FDR and accuracy improved to values of 22% and 76% respectively. We conclude that the CNN is not able to classify particles accurately in size bins where only few data of the considered class is available for training. However, the strong improvement after fine-tuning gives hope that this issue can be resolved after collecting more data and retraining the network.

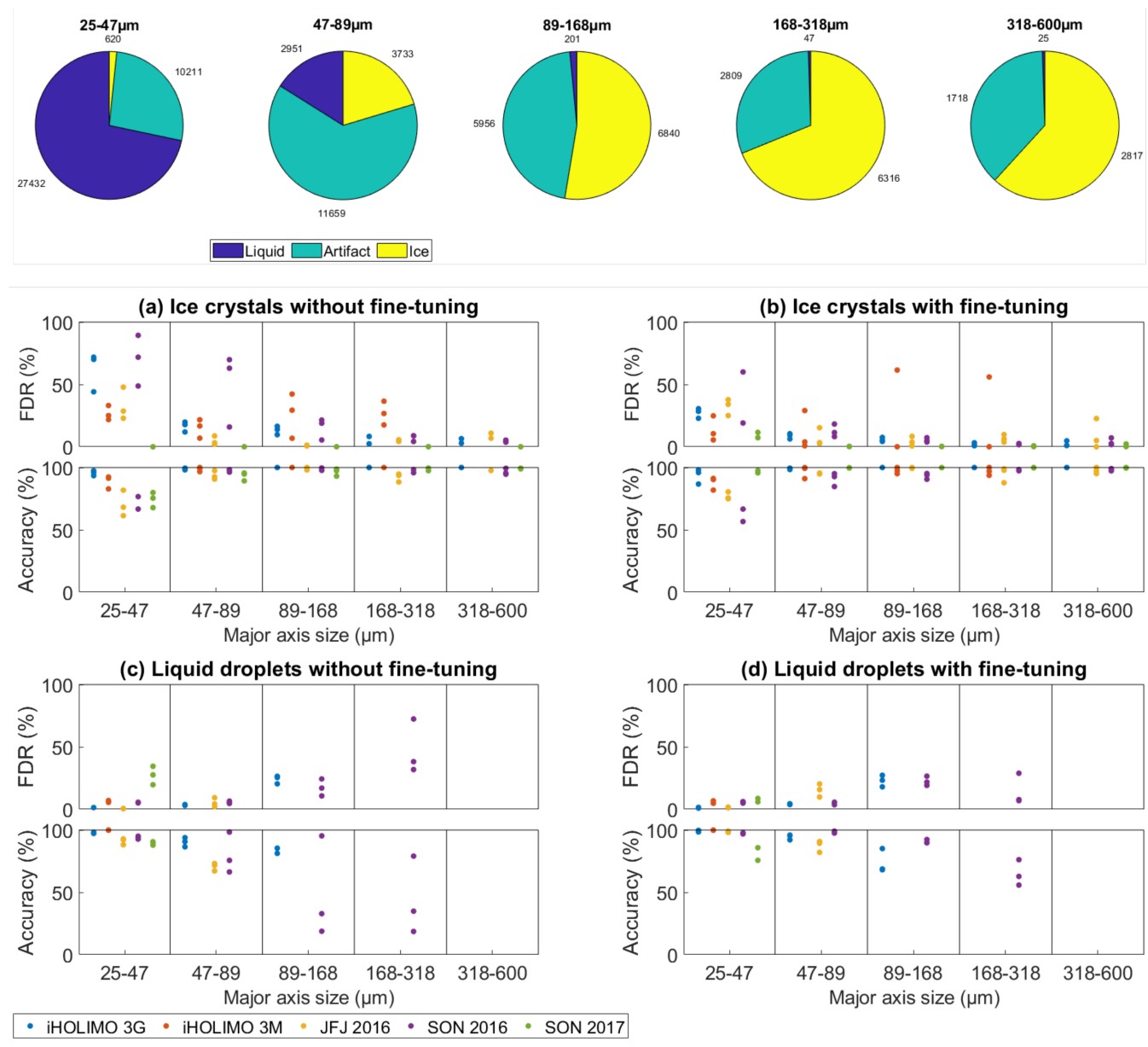

**Figure 12.** Evaluation of the CNN trained on four merged datasets and tested on the remaining fifth dataset for different particle sizes. The pie charts show the class distributions in the different size bins while the graphs show the accuracy and FDR results for ice crystals and liquid droplets before and after fine-tuning of three runs for the five datasets. Results of size bins where datasets have less than 30 particles of the considered class are excluded due to insufficient statistical representativeness.

## 5 Discussion

*Detecting mislabeled particles*

The data used for training the various classification algorithms presented above was hand-labeled and therefore prone to mislabeling. Mislabeling can happen either by crucial mistakes or because the truth class cannot be determined with certainty.

Mislabeling is more crucial in rare classes and/or size ranges.

Mislabeled particles which have unusual diameters for the class have a strong impact on the per-class accuracy or FDR in the corresponding size range and can, therefore, be detected as outliers (e.g. the orange dots in the top right of Fig. 12). After re-evaluation of the label of those samples, the CNN should be retrained. Uncertain cases should be excluded from the training set to not bias the CNN. Correcting the mislabeled particles and excluding the uncertain particles have the potential to improve

the prediction performance further.

*Comparison of the prediction performance of different classification algorithms*

Training and testing the CNN, SVM and the decision tree on the same single dataset led to similar accuracies and low FDRs in all cases (Fig. 9). Testing the same models on the other four datasets (generalization) all metrics worsened with the CNN

showing slightly better results than the decision tree and the SVM (Fig. 10). After merging four datasets and testing on a fifth unseen dataset, the CNN surpassed the results of the SVM and decision tree in almost every metric (Fig. 11). Therefore, we conclude that the CNN outperforms the decision tree and SVM in generalization tasks which is especially pronounced when a high quantity of diverse data was used for training.

Another important factor for the prediction performance is the time it takes to do the predictions. This highly depends on the

dataset and the computational power of the computer. Classifying 10,000 particles takes about 15 s for the decision tree, about 30 s for the SVM and about 60 s for the CNN on a local server. None of these time scales is comparable to the time it takes to classify 10,000 particles by hand, which can vary between a few hours and a few weeks depending on the dataset.

*Applying the CNN to new datasets*

We used the 15th and 85th percentile of the DGT of the three fine-tuning runs for all five datasets as an estimate of the lower and upper uncertainty. Thus, 70% of the data is included, which is similar to the definition of the standard error. The 50th percentile is the median and indicates a systematic bias. When considering the overall uncertainty an uncertainty of $\pm 10\%$ for ice crystals and $\pm 6\%$ for liquid droplets is sufficient to include most of the spread observed (see Table 2). For ice crystals smaller than 89 μm the uncertainty has to be set higher or a systematic correction should be applied, e.g. for ice crystals in the

first size bin about 15% and for the second size bin about 5% can be subtracted to the number of classified ice crystals with an uncertainty of $\pm 20\%$ for the first size bin and $\pm 10\%$ for the other size bins (see Table 3).

According to Fig. 2.5 in Beck (2017) the automated classification hitherto added about $\pm 10\%$ up to about +60% and -40% of uncertainty for particles smaller 40 μm. Other sources of uncertainty like the manual classification contribute with about $\pm 5\%$ (see Fig. 3) to the here considered size ranges. Therefore, the uncertainties using a fined-tuned CNN are of similar magnitude

**Table 2.** The 15th, 50th and 85th percentiles of the DGTs for ice crystals and liquid droplets of all sizes including all three runs (if available) for each of the five datasets after fine-tuning.

| Phase | 15th, 50th and 85th percentile of the DGT (%) |
|-------|-----------------------------------------------|
| Ice | -3 / 4 / 10 |
| Liquid | -1 / 1 / 6 |

**Table 3.** The 15th, 50th and 85th percentiles of the DGTs in % for ice crystals and liquid droplets for the different size bins including all three runs (if available) for each of the five datasets after fine-tuning.

| Size ranges | 25-47 µm | 47-89 µm | 89-168 µm | 168-318 µm | 318-600 µm |
|-------------|----------|----------|-----------|------------|------------|
| Ice | -4 / 18 / 27 | -3 / 4 / 12 | -3 / 2 / 8 | -4 / 1 / 6 | -1 / 1 / 6 |
| Liquid | -2 / 1 / 5 | -4 / 2 / 9 | - | - | - |

as uncertainties from other sources. Hence, the automated classification is no longer the main error source for the classification of small particles. However, the classification ability of the CNN cannot be assessed for droplets larger than 89 µm due to a lack of data in that size range.

*Needed accuracy of cloud particle classification regarding scientific questions*

How accurate the phase discrimination, the particle number or mass concentration has to be for a meaningful interpretation of the data highly depends on the scientific question. For example, in a model study, Young et al. (2017) showed that an overestimation of ICNC by only 17% ($2.43 \, 1^{-1}$ instead of $2.07 \, 1^{-1}$) led to cloud glaciation while the MPC was persistent for about 24h with the lower ICNC, while very few ice crystals ($0.21 \, 1^{-1}$ = -90%) may lead to cloud break-up. In theoretical

calculation, Korolev and Isaac (2003) showed that the glaciation time of a MPC with an ICNC of only $1 \, 1^{-1}$ is about four times as long as for $10 \, 1^{-1}$ (+100%) at a temperature of -15°C. Comparing measurements with studies can therefore already lead to wrong conclusions with classification uncertainties of ±20%.

## 6    Conclusion and Outlook

In this work, we evaluated the performance of a CNN for the classification task of images of ice crystals, liquid droplets and

artifacts of sizes between 25 µm and 600 µm from holographic imagers. The important takeaways from our experiments are summarized below.

*CNN compared to other machine learning baselines*

When training and testing the network on data from a single dataset, the CNN achieved similar results compared to a deci-

sion tree and an SVM. When training on a merged dataset and testing on an unseen dataset (generalization task), the CNN

surpassed the results of the decision tree and the SVM in six out of seven metrics, with the median overall accuracy being as high as 96.8%. Using the CNN not only improved the generalization ability but also requires less engineering effort since it automatically extracts features from data, unlike decision trees and SVMs.

### Fine-tuning

After fine-tuning the CNN trained on merged datasets with only 256 samples from the target dataset, the median of the overall accuracy improved from 96.8% to 98.2%. Similar improvements could be seen in all per-class accuracies and FDRs. Labeling of 256 samples from the target dataset is a very small amount of effort for improving the results considering the millions of samples acquired during a typical field excursion. However, the sample being used for fine-tuning has to be chosen carefully. All classes should be represented in similar amounts. Otherwise, fine-tuning can even worsen the results.

### CNN performance over particle size

Depending on the measurement parameter of interest (e.g. mass concentration), one might have to account for the variation of the performance ability of the CNN over particles size. Reanalysis should be done for liquid droplets larger than 89 µm in diameter and ice crystals smaller than 89 µm but also for sizes where the number of ice crystals exceeds the number of liquid droplets by at least an order of magnitudes and vice versa. We hope to mitigate this issue by collecting more data in these size ranges.

### Input channels

Using the phase images additionally to the amplitude images improved the results, even though we were not able to extract useful features of the phase images. This indicates that the CNN finds important features and might even exceed the classification abilities of humans.

### Usage for other imaging probes

The performance improvements seen after fine-tuning was applied to the CNN shows the ability of the CNN to adapt to different datasets. Therefore, we expect the CNN to also work for other imaging probes when fine-tuning is applied or after retraining the CNN from scratch. However, the phase channel has to be turned off since common imaging probes do not deliver phase images.

### Outlook

The use of the CNN to classify huge amounts of cloud particle data has a high potential to further constrain the phase-partitioning in MPCs. This will help to better understand the microphysical processes relevant for precipitation initiation and the glaciation of clouds. To improve the prediction of the CNN especially in for the classes unusual size bins, more data needs to be collected and labeled but also mislabeled particles should be relabeled or removed from the training dataset. Future plans also include adapting the CNN architecture presented in this paper to be capable of classifying particles smaller than 25 µm

and of discriminating different ice crystal habits. This will furthermore improve the analysis of especially small cloud particles and may help to answer important research questions about primary and secondary ice production.

*Code and data availability.*  All data and code is available at https://git.iac.ethz.ch/alauber/cnn_for_classifying_cloud_particles.git. The data for each figure can be found in .../Figures/Figure_XX. Check the README files for support.

## Appendix A: Neural Networks

The machine learning model used is a (deep) neural network which is used to discriminate between the three classes ice crystal, liquid droplet, and artifact. Deep neural networks are part of a broader family of techniques commonly referred to as deep learning. Over the past few years, deep learning has attracted a lot of attention in the machine learning community as well as in many other branches in science. One reason for this success is the high performance they have achieved on image classification (Krizhevsky et al., 2012). This success is to a large extent attributed to their ability to learn complex models while exhibiting good generalization properties. This means that these models are able to achieve good performance on new data that has not been seen while training the network.

A significant advantage of deep neural networks over more traditional machine learning models such as SVMs or decision trees is their ability to automatically learn useful data representations, thus eliminating the need for feature engineering.

In this work, we use Convolutional Neural Networks (CNNs) (LeCun et al., 1989), a type of neural networks particularly suitable for processing data that have a grid-like topology, such as images (grid of pixels). Here we give a brief introduction about CNNs and provide further details in the appendix.

### A1  Principal of a Neural Network

A neural network consists of an interconnected group of nodes named neurons. A neuron $n$ receives a $d$-dimensional vector $\mathbf{x}_n = (x_n^1, ..., x_n^d)$ as input and computes an output variable $o_n$ as

$$\boldsymbol{o_n} = f_n \left( \sum_{i=1}^{d} w_n^i \cdot x_n^i \right), \tag{A1}$$

where $\mathbf{w}_n = (w_n^1, ..., w_n^d)$ are the weight parameters of the neuron $n$ and $f_n$ is a chosen activation function. The weights $\mathbf{w}_n$ are trainable, meaning that they can be updated while training the network. The purpose of the activation function is to introduce a non-linearity to the otherwise linear model. Commonly used activation functions are the sigmoid, the hyperbolic tangent and the rectified linear unit (ReLU, Nair and Hinton (2010)) defined as $f(x) = max(0, x)$.

The neurons of a network are organized in collections called layers. A neural network consists of connected layers, effectively computing a function $f(\mathbf{x}; \boldsymbol{W}) = \hat{y}$, where $\mathbf{x}$ is the input image, $\boldsymbol{W}$ are the network parameters and $\hat{y}$ is the predicted label in a classification task setting.

### A2  Training a Neural Network

The parameters $\boldsymbol{W}$ of the network are learned in a supervised manner by using labeled data samples from the training dataset $\mathcal{X} = \{(\mathbf{x}_1, y_1), ..., (\mathbf{x}_T, y_T)\}$. Each training image $\mathbf{x}_t$ has a ground truth label $y_t$ associated to it. When $\mathbf{x}_t$ is used as input to the neural network, the label $f(\mathbf{x}_t; \boldsymbol{W}) = \hat{y}_t$ is predicted. We can define a loss function $L(y_t, \hat{y}_t)$ that measures the performance of the network for each data sample, compared to the ground truth. A commonly used loss function for multiclass classification

is softmax cross-entropy. The parameters $\boldsymbol{W}$ are learned by minimizing the loss

$$\boldsymbol{W}^* = \underset{\boldsymbol{W}}{\arg\min} \sum_{i=1}^{T} L(y_t, f(\mathbf{x}_t; \boldsymbol{W})).$$

This optimization problem is usually performed by using a gradient-based optimization method like stochastic gradient descent (SGD). Such methods require the calculation of the local gradients for the loss and the parameters of the network, which can be computed with the back-propagation algorithm, see details in Goodfellow et al. (2016).

## Appendix B: Pixel intensity distributions

An interesting aspect of the data are the pixel values for different classes and different datasets. We use the amplitude of the original complex images and compute the KDE (kernel density estimation) of the pixel intensity distribution for all the images of each class, for every dataset. The bandwidths of the kernels are computed using Scott's rule Scott (1979). The plots of the KDEs of the pixel intensity distributions are presented in Fig. B1.

We note that for all datasets the pixel intensity distributions of liquid droplets and ice crystals are bimodal. The peak at
the lower intensity (darker) corresponds to the pixels of the cloud particle shadow, whereas the peak at the higher intensity (brighter) corresponds to the pixels of the background that is included in the extracted images. The pixel intensity distribution for the artifacts has a single peak for all datasets. This is due to the fact that most artifacts are noisy images.

Another observation for all datasets is that the lower intensity peak is lower than the higher intensity peak for liquid droplets, while the opposite is true for ice crystals. This is mainly due to the smaller size on average of liquid droplets compared to ice
crystals. Since the background included in the extracted images is a fixed-size band of pixels around the border of the particle, this means that smaller cloud particles will have a larger percentage of their image covered by background pixels.

Aside from the aforementioned common characteristics, there are also some dissimilarities in the pixel intensity distributions for different datasets. First, the pixel intensities in datasets iHOLIMO 3G, SON 2017 and SON 2016 fall for the larger part in the $[0, 0.5]$ range, while those in datasets JFJ 2016 and iHOLIMO 3M seem to be scaled to the $[0, 0.6]$ range. Moreover,
in some cases the distribution for a class has differently shaped distributions for different datasets. The differences in the range and the shape of the pixel intensity distributions can be attributed to the use of different parameters for the automated particle reconstruction method, which affect the intensity values and the level of noise in the extracted images. Finally, the pixel intensity distributions for all images (whole dataset) vary according to the class frequencies of each dataset, since they can be considered as the weighted sum of the per-class distributions.

## Appendix C: Basic elements of a CNN

CNNs are primarily used with images as input. For this reason the neurons are arranged in three-dimensional structures $W \times H \times D$, where $W$ and $H$ are the width and height of the images and $D$ is their depth (number of channels).

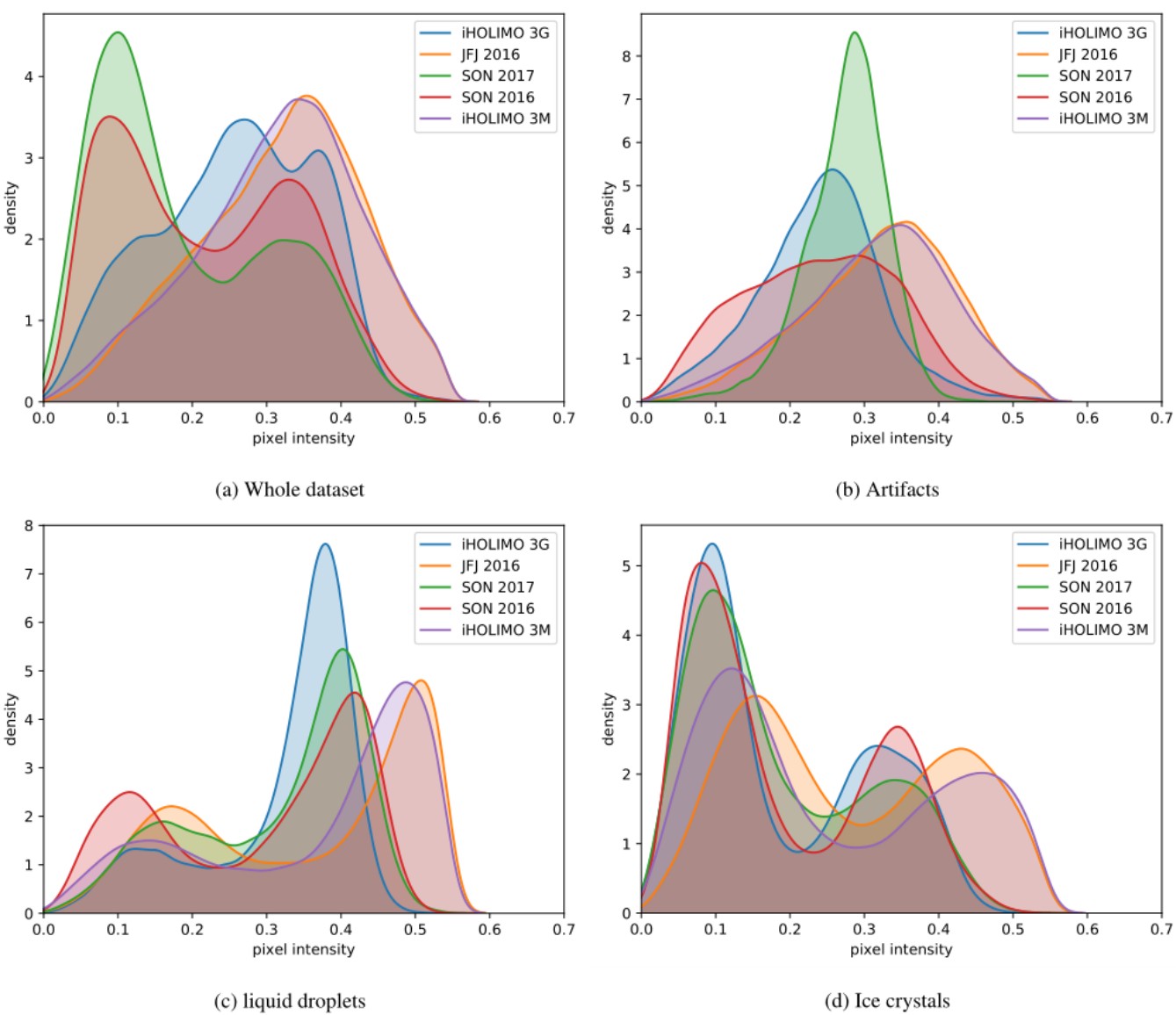

**Figure B1.** KDEs of the pixel intensity distributions for each class and for the whole dataset (all classes). The horizontal axis denotes the pixel intensity, from 0 for black to 1 for white.

A typical CNN architecture for image classification tasks consists of two parts. In the first part convolutional and pooling layers are repeatedly applied to extract features and reduce the spatial dimensions of the input. These features are the input to the second part of the network, where fully connected layers use them to perform the classification. The first part functions as a feature extractor and the second part as a classifier.

The three types of layers commonly used in CNNs are now briefly described.

## C1    Convolutional layers

Convolutional layers, the core building block of CNNs, transform an input 3D volume to an output 3D volume with potentially different dimensions.

Each convolutional layer is associated with sets of trainable weights called filters or kernels. A filter is a 3D structure 
$F \times F \times D$, where the spatial dimensions $F$ are small (typically 3, 5 or 7 pixels) and the depth dimension is equal to the depth of the input. Each filter is convolved over the input volume and computes dot products between its weights and the input at any position. The result of the convolution of one filter over the input is a 2D image. By choosing the number of the filters $K$ of the convolutional layer and stacking the computed 2D images, we end up with an output volume of depth $K$.

The sparse connectivity and parameter sharing properties of convolutional layers allow the network to learn filters that act 
similarly to feature detectors, detecting features present anywhere in the input image. The first layer of a deep CNN learns more general features such as edge detectors, while the deeper layers learn more complex and task specific features (Yosinski et al., 2014).

## C2    Pooling layers

The other key component of convolutional networks is the pooling layer. The purpose of this layer is to reduce the spatial 
dimensions of the intermediate convolutional layers of a deep CNN, while leaving the depth dimension unchanged. This is achieved by performing a down sampling operation on every depth slice of the input. A commonly used configuration is max pooling (Zhou and Chellappa, 1988) with stride $S = 2$, where each depth slice of the input is downs ampled by two along width and height, by keeping the maximum value in $2 \times 2$ non-overlapping regions of the input.

## C3    Fully connected layers

In a fully connected layer, each neuron accepts as input the output of every neuron from the previous layer. Fully connected layers perform the high-level reasoning in a CNN after the convolutional and pooling layers. For classification tasks, the last fully connected layer has as many neurons as the classes and each one represents the corresponding class score. Softmax is usually applied to transform the class scores into class probabilities. To make a prediction for one data point fed to the network, the class with the highest probability is chosen.

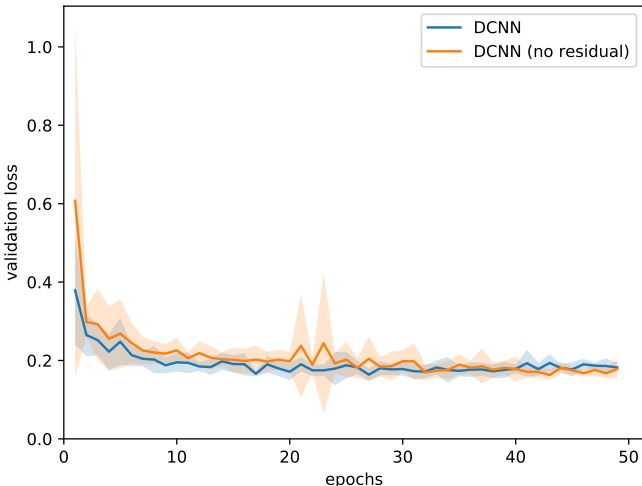

**Figure D1.** Validation loss during training for different models. Average loss and standard deviation over 10 repetitions of the merged dataset experiment for iHOLIMO 3G.

### Appendix D: Training

The validation losses during training for training the DCNN model with and without dense residual connections, is presented in Fig. D1. We can see that when using dense residual connections, the validation loss decreases faster.

### Appendix E: Misclassified particles

In order to investigate the cases where the CNN fails to classify the images correctly, we visualize samples of the misclassified images for all class pairs. In Table E1 we have misclassified samples for the CNN model trained on a merged dataset and evaluated on the SON 2016 dataset.

We observe that some images labeled as artifacts are in fact liquid droplets and the CNN classifies them correctly as such. In some cases, other images are also incorrectly labeled as ice crystals or liquid droplets. Some liquid droplets are cropped irreg-
ularly by the particle reconstruction algorithm and the CNN confuses them with artifacts. Finally, large-sized liquid droplets are sometimes incorrectly classified as ice crystals, because the network does not learn to classify them correctly due to their rarity.

### Appendix F: t-SNE

To investigate how the different models adapt to the target domain, we visualize the features of the second-to-last fully con-
nected layer for the source and target dataset using t-SNE in Fig. F1. t-SNE is method for dimensionality reduction for em-

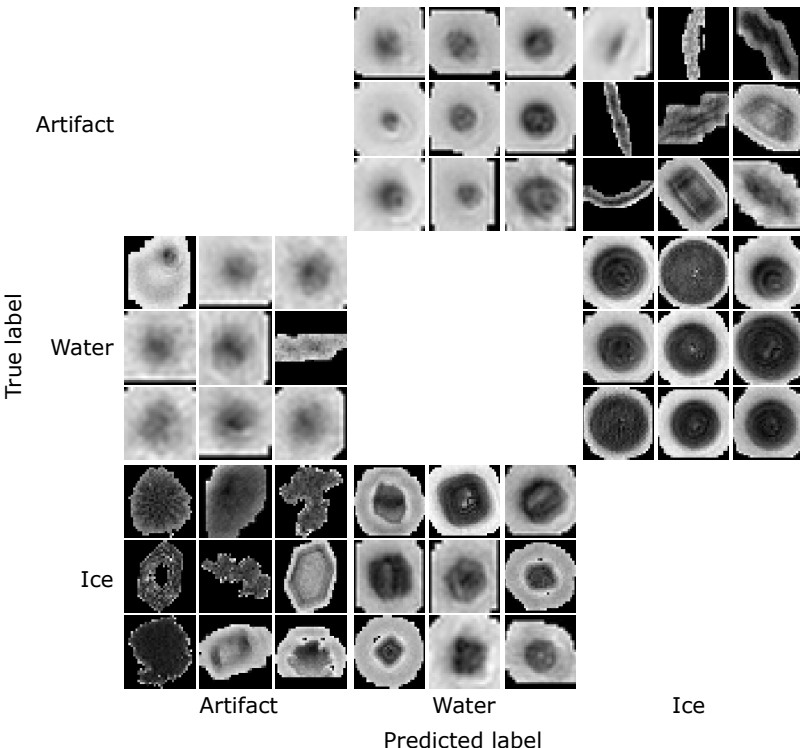

**Figure E1.** Misclassified samples from the evaluation of the CNN model, trained using a merged dataset, on the SON 2016 dataset. Visualized using amplitude images.

bedding high-dimensional data into a 2D space, in such a way that similar inputs are modeled by nearby points and dissimilar inputs are modeled by distant points. For an effective domain adaptation model, the features for different classes must be easily discriminable and the features for the same class must align for the source and target datasets.

In Fig. F1 (a) and (c) we observe that CNN already leads to good class separability, which explains the good results. There is also good alignment between the domains, except for the liquid droplet embeddings. There are two liquid droplet clusters in the source domain plot, but only one in the target. This is because the merged dataset contains data different distributions, including the target domain distribution.

Fine-tuning improves the class separability in the target domain, but worsens it in the source as we can see in Fig. F1 (b) and (d). This is because fine-tuning results in a model specialized on the target dataset, while the performance on the source domain is not taken into consideration.

**Table G1.** Description of all features used for training and testing the decision tree and SVM. More detailed descriptions of the features can be found in Schlenczek (2018).

| Metric | Meaning |
| --- | --- |
| area | area |
| asprat | aspect ratio |
| eqsiz | mean diameter of major and minor axis |
| majsiz | length of particle major axis |
| maxampg | maximum amplitude gradient in particle-patch |
| maxcompg | maximum complex gradient in particle-patch |
| maxph | maximum phase in particle-patch |
| maxphg | maximum phase gradient in particle-patch |
| meanampg | mean amplitude gradient in particle-patch |
| meancompg | mean complex gradient in particle-patch |
| meanph | mean phase in particle-patch |
| meanphg | mean phase gradient in particle-patch |
| minampg | minimum amplitude gradient in particle-patch |
| mincompg | minimum complex gradient in particle-patch |
| minph | minimum phase in particle-patch |
| minphg | minimum phase gradient in particle-patch |
| minsiz | length of particle minor axis |
| numzs | z-depth of chain of this particle |
| orient | particle orientation |
| pampdepth | depth of particle amplitude relative to background amplitude |
| phfl | phase flip |
| rngamp | range of amplitude in particle-patch |
| rngampg | range of amplitude gradient in particle-patch |
| rngcompg | range of complex gradient in particle-patch |
| rngph | range of phase in particle-patch |
| rngphg | range of phase gradient in particle-patch |
| stdamp | standard deviation of amplitude in particle-patch |
| stdampg | standard deviation of amplitude gradient in particle-patch |
| stdcompg | standard deviation of complex gradient in particle-patch |
| stdph | standard deviation of phase in particle-patch |
| stdphg | standard deviation of phase gradient in particle-patch |
| underthresh | underthresh |

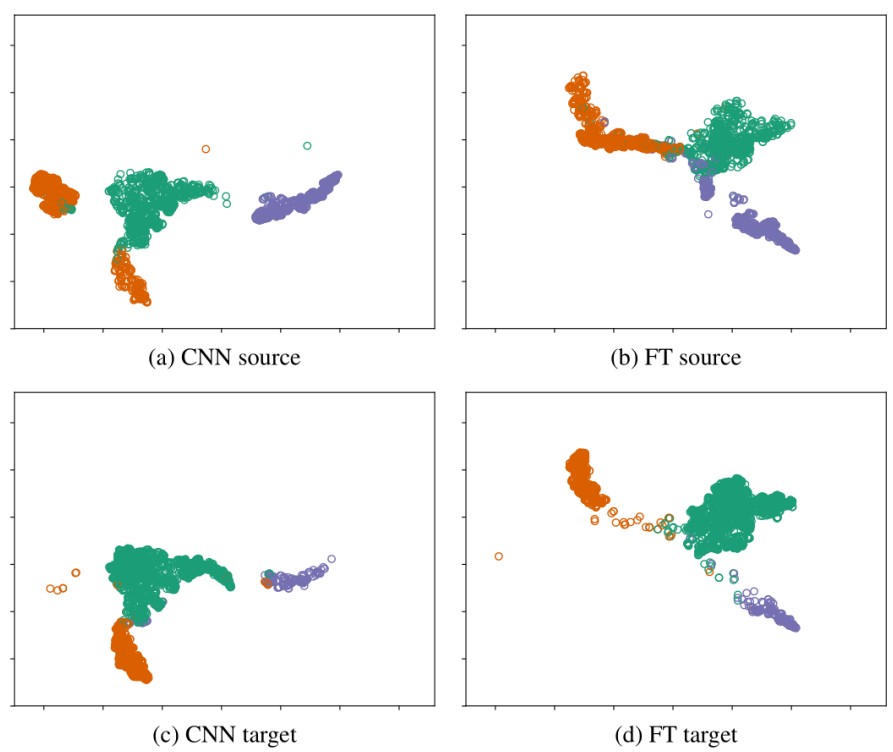

(a) CNN source  (b) FT source

(c) CNN target  (d) FT target

**Figure F1.** t-SNE plots of the features of the second-to-last fully connected layer for the merged dataset experiments for JFJ 2016 using the different models. Green: Artifacts, Orange: liquid droplets, Purple: Ice crystals. The same embedding space and perplexity 30 was used for all plots.

## Appendix G: Features

*Author contributions.* Georgios Touloupas designed and developed the CNN architecture with main inputs regarding the machine learning part by Aurélien Lucchi. Georgios Touloupas also performed all used runs with the CNN. Annika Lauber provided the training data, did the decision tree and SVM runs and the interpretation of the results. The manuscript was mainly written by Annika Lauber and partly by Georgios Touloupas. All authors discussed the results and commented on the manuscript.

*Competing interests.* The authors declare that they have no conflict of interest.

*Acknowledgements.* First of all, the authors like to thank Darrel Baumgardner and the second anonymous referee for their valuable feedbacks on our manuscript. We thank the whole Holosuite team, especially Aaron Bansemer, Maximilian Weitzel, Neel Desai, Susanne Glienke, Sarah Barr and Fabiola Ramelli for providing us with labeled data and/or information about their data analysis procedure. We also want to thank the Atmospheric Physics group at ETH-IAC for feedback and discussion, especially Claudia Marcolli and David Neubauer for comments on the paper draft. This project was supported by the SNF grant 200021_175824.

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
