# Peer review of "A Convolutional Neural Network for Classifying Cloud Particles Recorded by Imaging Probes"

_Atmospheric Measurement Techniques, 2019_

## Referee Comment (RC1) · Darrel Baumgardner (Referee) · 10 Aug 2019

The approach of using the convolutional neural network (CNN) for processing images from cloud imaging probes can certainly be considered novel, given that it has not been attempted prior to this study. What puzzles me, however, is why it is being put to such mundane use as only determining if an image is circular or not, given that there are a number of less complicated and less compute intensive approaches that have already been tested and implemented in the literature. What would have been truly novel, given that they actually show dendritic crystals in the example, would be to show how the CNN could identify different crystal habits, not just separate circular from non-circular.

[Figure]

I do not think that this analysis approach will be widely implemented by the cloud physics community at large, given the existence of much simpler, faster, and likely as effective methods to separate circular from non-circular. The majority of cloud physicists that use cloud probes are not currently working with holography and are unlikely to consider the CNN since only holographic examples are used in this paper.

That being said, although I think this submission is much less interesting due to the mundane application, I won't reject it on those grounds. There are, however, a number of additions that I think are necessary to include before I would accept this paper for publication.

1) How are holographic images currently being processed to separate ice from droplets? If other than those machine learning references given in the paper, then they need to be discussed and compared with the CNN.

2) What are the techniques that are currently being used to separate ice from droplets in other imaging systems like optical array probes (OAPs)?

3) If the techniques being used for processing images in OAPs can be used in holographic images (I can see no reason why they can't given the rendering of holographic images into 2D for the CNN) how do the error rates compare with those from CNN?

4) Nowhere in the introduction, or elsewhere, do the authors discuss how errors in discriminating liquid from ice will impact how measurements are interpreted with respect to scientific questions associated with mixed phase conditions. This is a critical omission when assessing the efficiency of one technique versus the other. Particularly when it comes to ease of implementing one technique versus the other. Creating training sets for every data set is time consuming and one that is unnecessary if using one of the common techniques used in OAP analysis.

5) Why hasn't the uncertainty in human-typing images been assessed, i.e. having at least two or more observers classify the same data set? Isn't that a fundamental step

when determining how well any automatic classification scheme performs?

6) In addition to the number of hours needed to create a training set, what are the computational times to analyze sample data sets of 10000 images by each technique?

7) The list of references is missing a large number of studies on pattern recognition of cloud probe images. These have to be included.

---

## Referee Comment (RC2) · Anonymous Referee #2 · 11 Aug 2019

"A Convolutional Neural Network for Classifying Cloud Particles Recorded by Imaging Probes" describes the application of a convolutional neural network for classification of images of atmospheric hydrometeors from holographic imagers and other imaging probes. This technique could be of great help in particular for the classification of holographic data sets. This topic is of interest for the atmospheric sciences community and fits the scope of AMT. The application of the CNN is explained in great detail and the text is well-written but there are various things that should be added to the manuscript prior to publication.

General comments:

(1) The introduction needs more depth in terms of methods suitable for ice versus liquid water particle classification. Here, particle classification by shape is discussed in great

detail but there are also other possibilities, e.g. using the forward scattering pattern as it is done by the SID-3 instrument. Please discuss also the other options to distinguish ice from liquid water in the introduction. There are examples given in Korolev et al. (2017) and other references.

(2) There is way more literature about ice particle classification available. For example, O'Shea et al. (2016) were also using a machine learning approach to classify ice particles with the distinction between spheres and other shapes being part of it. I would like to see a comparison of the different schemes in the introduction, and a clearer presentation of what is the innovative part of your method in this context.

(3) Images of droplets obtained by a holographic imager tend to appear somewhat distorted at far away distance from the camera and the lateral center of the detector. Could that be one reason for a rather low accuracy in the classification of small ice crystals?

(4) Would the skill of the CNN improve if out-of-focus particles (a common type of objects that can be seen as artifact) were removed completely from the sample and processed separately with different methods?

Specific comments:

Page 2, line 10: The minimum diameter issue is also mentioned in Korolev et al. (2017) - it is Figure 5-10 that demonstrates this issue very well.

Page 3, line 23: At this point it is useful to discuss the accuracy of the focus finder subroutine. How much can a particle be out of focus to still be recognizable as a particle and not being treated as an artifact?

Figure 2: It would be helpful to get an idea of the phase and amplitude values and the particle size (how much stronger is the amplitude signal of the particle in comparison to background noise?)

Page 4, line 3: Do you have an estimate of how strong the classification bias typically

is? One way to estimate the classification bias is to classify the same dataset by two or more independent users.

Page 6, line 7: Please provide some evidence that this is really an optimum choice (either in supplement or another appendix).

Page 6, line 9: How is the zero padding interpreted by the CNN? This is something that could introduce a bias.

Page S1, line 31: What does KDE stand for?

References:

Korolev, A., G. McFarquhar, P. R. Field, C. Franklin, P. Lawson, Z. Wang, E. Williams, S. Abel, D. Axisa, S. Borrmann, J. Crosier, J. Fugal, M. Krämer, U. Lohmann, O. Schlenczek and M. Wendisch (2017): Chapter 5: Mixed-phase clouds: progress and challenges. AMS Meteorological Monographs 58, 5.1-5.50.

O'Shea, S. J., T. W. Choularton, G. Lloyd, J. Crosier, K. N. Bower, M. Gallagher, S. J. Abel, R. J. Cotton, P. R. A. Brown, J. P. Fugal, O. Schlenczek, S. Borrmann and J. C. Pickering (2016): Airborne observations of the microphysical structure of two contrasting cirrus clouds. J. Geophys. Res.: Atmospheres 121 (22), 13510-13536.2016JD025278.

---

## Author Comment (AC1) · 11 Nov 2019

**Response to referee 1**

First of all, we like to thank Darrel Baumgardner for his thorough and valuable feedback on our paper. In the following, we will address all his comments and show the according changes we made on the paper.

**Comment 1:**

How are holographic images currently being processed to separate ice from droplets? If other than those machine learning references given in the paper, then they need to be discussed and compared with the CNN.

**Answer to comment 1:**

We are not aware of any other machine learning technique apart from SVMs and decision trees being used by the holography community to classify holographic images.

**Comment 2:**

What are the techniques that are currently being used to separate ice from droplets in other imaging systems like optical array probes (OAPs)?

**Answer to comment 2:**

There are numerous approaches to separate ice particles from liquid droplets. To acknowledge them we added line 26 to line 31 on page 2 in the introduction:

"Imaging probes, which differentiate only ice from liquid usually extract features from the images that measure the circularity of the particles (e.g. Korolev and Sussman (2000); Crosier et al. (2011); Lawson et al. (2001)). Korolev and Sussman (2000) state an uncertainty for differentiating spheres from irregular particles of 20% to 25% for a pixel number between 20 and 60 and a few percents for higher pixel numbers. These values are comparable to our results for holographic images, which we will introduce later. However, the existing approaches are not suitable for holographic images since they do not account for artifacts. Finding good features for artifacts is difficult because they do not have a specific shape."

**Comment 3:**

If the techniques being used for processing images in OAPs can be used in holographic images (I can see no reason why they can't given the rendering of holographic images into 2D for the CNN), how do the error rates compare with those from CNN?

**Answer to comment 3:**

As already described in the answer to comment 2, we cannot use these algorithms due to the existence of artifacts in our datasets. Most of the mentioned studies do not give an uncertainty estimation and if they do, it is not easily possible to compare it to our results since they consider different sizes of particles than we do and have different pixel size.

**Comment 4:**

Nowhere in the introduction, or elsewhere, do the authors discuss how errors in discriminating liquid from ice will impact how measurements are interpreted with respect to scientific questions associated with mixed phase conditions. This is a critical omission when assessing the efficiency of one technique versus the other. Particularly when it comes to ease of implementing one technique versus the other.

Creating training sets for every data set is time consuming and one that is unnecessary if using one of the common techniques used in OPA analysis.

**Answer to comment 4:**

We added the section "Needed accuracy of cloud particle classification regarding scientific question" into the discussion section (line 5 to line 12 on page 20):

"Needed accuracy of cloud particle classification regarding scientific questions. How accurate the phase discrimination, the particle number or mass concentration has to be for a meaningful interpretation of the data highly depends on the scientific question. For example, in a model study, Young et al. (2017) showed that an overestimation of ICNC by only 17% (2.43 $l^{-1}$ instead of 2.07 $l^{-1}$) led to cloud glaciation while the MPC was persistent for about 24h with the lower ICNC, while very few ice crystals (0.21 $l^{-1}$ = -90%) may lead to cloud break-up. In theoretical calculation, Korolev and Isaac (2003) showed that the glaciation time of a MPC with an ICNC of only 1 $l^{-1}$ is about four times as long as for 10 $l^{-1}$ (+100%) at a temperature of -15°C. Comparing measurements with studies can therefore already lead to wrong conclusions with classification uncertainties of ±20%."

**Comment 5:**

Why hasn't the uncertainty in human-typing images been assessed, i.e. having at least two or more observers classify the same data set?

**Answer to comment 5:**

The human bias is now assessed with the results of three people classifiying the same dataset. To show our results we added line 11 to 16 on page 5 in the Experimental data section together with Figure 3. We also adapted line 33 on page 19 to line 2 on page 20 of the Discussion section "Applying the CNN to new datasets" accordingly.

"For the estimation of the human bias, three different people hand-labeled the same dataset consisting of 1000 particles. The number of particles hand-labeled as the considered class by at least one person are compared to the number of particles hand-labeled as the considered class by all three persons. Taking the average of these two numbers, the spread can be given as the percentage deviation to the two values (see Fig. 3). For liquid droplets, we have a deviation of ±4% and for ice crystal ±5%. However, this estimation does not take into account that in some cases humans might just not be able to recognize the correct class as outlined before."

"Other sources of uncertainty like the manual classification contribute with about ±5% (see Fig. 3) to the here considered size ranges. Therefore, the uncertainties using a fined-tuned CNN are of similar magnitude as uncertainties from other sources"

**Comment 6:**

In addition to the number of hours needed to create a training set, what are the computational times to analyze sample data sets of 10000 images by each technique?

**Answer to comment 6:**

We added line 19 to 22 on page 19 in the discussion section:

"Another important factor for the prediction performance is the time it takes to do the predictions. This highly depends on the dataset and the computational power of the computer. Classifying 10,000 particles takes about 15 s for the decision tree, about 30 s for the SVM and about 60 s for the CNN on a local server. None of these time scales is comparable to the time it takes to classify 10,000 particles by hand, which can vary between a few hours and a few weeks depending on the dataset."

**Comment 7:**

The list of references is missing a large number of studies on pattern recognition of cloud probe images. These have to be included.

**Answer to comment 7:**

To include studies on cloud pattern recognition of cloud probe images we added line 26 to line 31 on page 2 (the section is already written-out in the answer to comment 2) as well as line 8 to 14 on page 3 in the introduction:

"Deep learning (usually referred to as neural networks) has the potential to overcome transfer learning issues, which we will show in this work. For the classification of cloud particles, a feedforward neural network from Hagan and Menhaj (1994) was used by O'Shea et al. (2016) to classify CPI data into different ice particle shapes and liquid droplets. The network is fed by different features, which are calculated beforehand. Their results are promising with a total accuracy of 88% to classify the images into six habits including liquid droplets for particles larger than 50 μm. This type of a neural network also requires feature extraction and does not work for holographic images because it does not account for a class without a specific shape like artifacts."

**Additional comment by authors about a change of a used metric**

We want to point out that we changed the used metric "equivalent area particle diameter" for the evaluation of the CNN on different particle sizes to "major axis size". The reason for this is that we noticed that the equivalent area particle diameter was not calculated correctly. We, therefore, decided to use the major axis size instead, which is also a measure of size. The changes in the results are small and the interpretation of the results does not change.

---

## Author Comment (AC2) · 11 Nov 2019

**Response to referee 2**

First of all, we like to thank the anonymous referee for his/her thorough and valuable feedback on our paper. In the following, we will address all comments and show the according changes we made on the paper.

**General comment 1:**

The introduction needs more depth in terms of methods suitable for ice vs. liquid water particle classification. Here, particle classification by shape is discussed in great detail but there are also other possibilities, e.g. using the forward scattering pattern as it is done by the SID-3 instrument. Please discuss also the other options to distinguish ice from liquid water in the introduction. There are examples given in Korolev et al. 2017 and other references.

**Answer to general comment 1:**

To also introduce other instruments, which are suitable for ice vs. liquid measurements, we added line 23 on page 1 to line 6 on page 2 in the introduction:

"For single-particle detection, there are two common measurement instruments: imaging and light scattering probes. The latter (e.g. SID, Cotton et al. (2010), BCP, Beswick et al. (2014), CAS, Baumgardner et al. (2001)) capture the scattered light of a single particle usually over a range of angles. Applying Mie theory and scale factors, which are derived from calibrations, information of the measured particle like the equivalent optical diameter (EOD) can be derived. However, this can be a major issue for nonspherical ice crystals since the derivation of the EOD assumes sphericity and the exact shape of the captured particle is unknown (Baumgardner et al., 2017).

This issue is partly overcome with imaging probes, which capture images of the particle itself. Assumptions of the shape have only to be made on the third dimension and if the resolution is low compared to the particle size like outlined later in this section."

**General comment 2:**

There is way more literature about ice particle classification available. For example, O'Shea et al (2016) were also using a machine learning approach to classify ice particles with the distinction between spheres and other shapes being part of it. I would like to see a comparison of the different schemes in the introduction and a clearer presentation of what is the innovative part of your method in this context.

**Answer to general comment 2:**

To include more literature dealing with cloud particle classification, we added line 26 to line 31 on page 2 as well as line 8 to line 14 on page 3 in the introduction:

"Imaging probes, which differentiate only ice from liquid, usually extract features from the images that measure the circularity of the particles (e.g. Korolev and Sussman (2000); Crosier et al. (2011); Lawson et al. (2001)). Korolev and Sussman (2000) state an uncertainty for differentiating spheres from irregular particles of 20% to 25% for a pixel number between 20 and 60 and a few percent for higher pixel numbers. This values are comparable to our results for holographic images, which we will introduce later. However, the existing approaches are not suitable for holographic images since they do not account for artifacts. Finding good features for artifacts is difficult because they do not have a specific shape."

"Deep learning (usually referred to as neural networks) has the potential to overcome transfer learning issues, which we will show in this work. For the classification of cloud particles, a feedforward neural

network from Hagan and Menhaj (1994) was used by O'Shea et al. (2016) to classify CPI data into different ice particle shapes and liquid droplets. The network is fed by different features, which are calculated beforehand. Their results are promising with a total accuracy of 88% to classify the images into six habits including liquid droplets for particles larger than 50 µm. This type of a neural network also requires feature extraction and does not work for holographic images because it does not account for a class without a specific shape like artifacts."

**General comment 3:**

Images of droplets obtained by a holographic imager tend to appear somewhat distorted at far away distance from the camera and the lateral center of the detector. Could that be one reason for a rather low accuracy in the classification of small ice crystals?

**Answer to general comment 3:**

To assess this question, we plotted the accuracy and FDR in all three directions for particles with a major axis size smaller than 47 µm in Figure 1. No trend towards lower accuracies at the edges is obvious.

In short: yes, images of droplets tend to appear somewhat distorted at far away distance from the camera. However, this does not seem to be the reason for a rather low accuracy in the classification of small ice crystals.

[Figure]

*Figure 1: The accuracy and FDR values for different bins in the horizontal (x), the vertical (y) and perpendicular to the camera (z) direction of the same runs as shown in Figure 11 in the main manuscript without fine-tuning being applied. Missing datapoints of datasets in position bins are due to different considered volumes of the datasets.*

**General comment 4:**

Would the skill of the CNN improve if out-of-focus particles (a common type of objects that can be seen as artifact) were removed completely from the sample and processed separately with different methods?

**Answer to general comment 4:**

The skill of the CNN would probably be improved if out-of-focus particles were removed. However, we are not aware of out-of-focus particles being part of the datasets. We usually don't have a big issue with out-of-focus particles.

**Specific comment 1:**

Page 2, line 10: The minimum diameter issue is also mentioned in Korolev et al. (2017) – it is Figure 5-10 that demonstrates this issue very well.

**Answer to specific comment 1:**

We adapted line 16 to line 18 on page 2 to also give credit to Korolev et al. (2017):

"To identify needles six pixels might be enough, whereas a minimum of 12 to 15 pixels is required to identify plates while frozen droplets which are spherical cannot be detected regardless of their diameter (Korolev and Sussman, 2000; Korolev et al., 2017)."

**Specific comment 2:**

Page 3, line 23: Discuss the accuracy of the focus finder subroutine. How much can a particle be out of focus to still be recognizable as a particle and not being treated as an artifact?

**Answer to specific comment 2:**

We reconstructed holograms in about every 100 µm. For particles of the considered sizes, this is enough to differentiate artifacts from particles because even particles of a major axis of 25µm are still recognizable as particles in the neighboring planes. More details in Schlenczek (2018) [1], chapter 4.1.4.

**Specific comment 3:**

Figure 2: It would be helpful to get an idea of the phase and amplitude values and the particle size (how much stronger is the amplitude signal of the particle in comparison to background noise?)

**Answer to specific comment 3:**

We decided that the shown phase image in the figure was misleading since it did not show a typical phase image and we, therefore, changed it. How much stronger the amplitude signal of a particle is in comparison to the background noise depends strongly on the quality of the dataset and the used threshold at which particles are still recognized by the software and can therefore not be given as a number. However, we hope that the new figure gives a better impression of typical amplitude and phase images. We also added a size scale to get an impression of the particle size.

**Specific comment 4:**

Page 4, line 3: Do you have an estimate of how strong the classification bias typically is? One way to estimate the classification bias is to classify the same dataset by two or more users.

**Answer to specific comment 4:**

The human bias is now assessed with the results of three people classifying the same dataset. To show our results we added line 11 to 16 on page 5 in the Experimental data section together with Figure 3. We also adapted line 33 on page 19 to line 2 on page 20 of the Discussion section "Applying the CNN to new datasets" accordingly.

"For the estimation of the human bias, three different people hand-labeled the same dataset consisting of 1000 particles. The number of particles hand-labelled as the considered class by at least one person are compared to the number of particles hand-labeled as the considered class by all three persons. Taking the average of these two numbers, the spread can be given as the percentage deviation to the two values (see Fig. 3). For liquid droplets, we have a deviation of ±4% and for ice crystal ±5%. However, this estimation does not take into account that in some cases humans might just not be able to recognize the correct class as outlined before."

"Other sources of uncertainty like the manual classification contribute with about ±5% (see Fig. 3) to the here considered size ranges. Therefore, the uncertainties using a fined-tuned CNN are of similar magnitude as uncertainties from other sources"

**Specific comment 5:**

Page 6, line 7: Please provide some evidence that this is really an optimum choice.

**Answer to specific comment 5:**

We do not claim that 32x32 is the optimal image size, but that we empirically found that this leads to satisfying results. Using a 64x64 image size did not lead to a significant improvement in the performance of the CNN.

**Specific comment 6:**

How is the zero padding interpreted by the CNN? This is something that could introduce a bias.

**Answer to specific comment 6:**

We use zero-padding as this is the most widely adopted padding scheme for deep learning, due to its simplicity and computational efficiency. The goal is for the CNN to learn to interpret it as the border of the full particle image. Padding with the mean of the image or dataset pixel intensity did not improve the results. (Liu et al. 2018) [2]

**Specific comment 7:**

Page S1, line 31: What does KDE stand for?

**Answer to specific comment 7:**

KDE stands for kernel density estimation, which is a way to estimate a probability density function. It was included in the text.

**Additional comment by authors about a change of a used metric**

We want to point out that we changed the used metric "equivalent area particle diameter" for the evaluation of the CNN on different particle sizes to "major axis size". The reason for this is that we noticed that the equivalent area particle diameter was not calculated correctly. We, therefore,

decided to use the major axis size instead, which is also a measure of size. The changes in the results are small and the interpretation of the results does not change.

**References**

1.  Schlenczek O. Airborne and Ground-based Holographic Measurement of Hydrometeors in Liquid-phase, Mixed-phase and Ice Clouds. 2018.

2.  Guilin Liu, Kevin J. Shih, Ting-Chun Wang, Fitsum A. Reda, Karan Sapra, Zhiding Yu, Andrew Tao BC. *Partial Convolution Based Padding*.; 2018.